



# Impact of coastal East Antarctic ice rises on surface mass balance: insights from observations and modeling

Thore Kausch[1], Stef Lhermitte[1], Jan T.M. Lenaerts[2], Nander Wever[2], Mana Inoue[3], Frank Pattyn[3], Sainan Sun[3], Sarah Wauthy[3], Jean-Louis Tison[3], and Willem Jan van de Berg[4]

[1]Delft University of Technology, Mekelweg 5, 2628 CD Delft, The Netherlands
[2]Department of Atmospheric and Oceanic Sciences, University of Colorado Boulder, 4001 Discovery Dr., CO 80309 Boulder, The United States of America
[3]Université Libre de Bruxelles, Avenue F.D. Roosevelt 50, B-1050 Bruxelles, Belgium
[4]Utrecht University, Princetonplein 5, 3584 CC Utrecht, The Netherlands

**Correspondence:** Thore Kausch (t.kausch@tudelft.nl)

**Abstract.** About 20 % of all snow accumulation in Antarctica occurs on the ice shelves. There, ice rises control the spatial surface mass balance (SMB) distribution by inducing snowfall variability and wind erosion due to their topography. Moreover these ice rises buttress the ice flow and represent ideal drilling locations for ice cores. In this study we assess the connection between snowfall variability and wind erosion to provide a better understanding of how ice rises impact SMB variability, how well this is captured in the regional atmospheric climate model RACMO2, and the implications of this SMB variability for ice rises as an ice core drilling site. By combining ground penetrating radar (GPR) profiles from two ice rises in Dronning Maud Land with ice core dating we reconstruct spatial and temporal SMB variations from 1982 to 2017 and compare the observed SMB with output from RACMO2 and SnowModel. Our results show snowfall driven differences of up to 1.5 times higher SMB on the windward side of both ice rises than on the leeward side, as well as a local erosion driven minimum at the ice divide of the ice rises. RACMO2 captures the snowfall driven differences, but overestimates their magnitude, whereas the erosion on the peak can be reproduced by SnowModel with RACMO2 forcing. Observed temporal variability of the average SMBs, retrieved from the GPR data for four time intervals in the 1982-2017 range, are low at the peak of the easternmost ice rise (~ 0.03 mw.e./yr), while being three times higher (~ 0.1 mw.e./yr) on the windward side of the ice rise. This implies that at the peak of the ice rise, higher snowfall, driven by orographic uplift, is balanced out by local erosion. As a consequence of this the SMB recovered from the ice core matches the SMB from the GPR at the peak of the ice rise, but not at the windward side of the ice rise, suggesting that the SMB signal is dampened in the ice core.

*Copyright statement.* TEXT

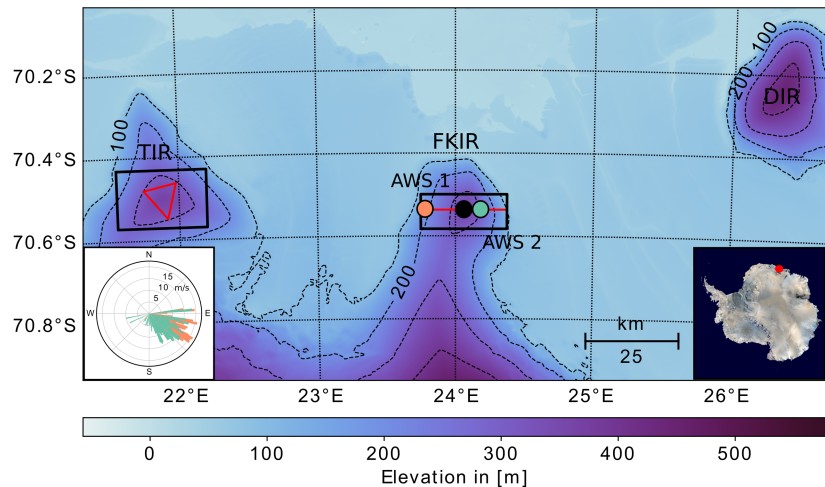

**Figure 1.** Topography of the ice rise and the surrounding area in Dronning Maud Land, based on the 90m TanDEM-X digital elevation model (© DLR 2018). The red lines represent the GPR profiles, the black boxes the areas on which Snowmodel was applied and the two orange and green dots mark the locations of two AWS installed on the eastern and western flank of the FKIR. The bar plot in the lower left corner shows the wind speed and direction measured by the two AWS between January 2018 and April 2018, where orange and green denote AWS 1 and 2, respectively.

## 1 Introduction

The surface mass balance (SMB) remains one of the largest sources of uncertainty when determining the overall mass balance
of Antarctica and with that Antarctica's contribution to sea level rise (Lenaerts et al., 2019; The IMBIE team, 2018; Rignot et al., 2019). The SMB of an ice sheet is commonly defined as the annual sum of all surface processes that affect the mass balance of an ice sheet. This includes snowfall, runoff of surface melt water, sublimation, drifting snow sublimation, as well as snow erosion and deposition, of which snowfall is the dominant component in East Antarctica (Boening et al., 2012). The SMB varies substantially in space across East Antarctica, with the highest SMB found on the coastal margins of the ice sheet and
very low SMB on the East Antarctic Plateau, where the air is too cold to hold large amounts of moisture (Vaughan et al., 1999; Rotschky et al., 2007; Monaghan et al., 2006). The coastal regions of the East Antarctic ice sheet are not only characterized by high SMB, but also by substantial inter annual as well as spatial variability (Lenaerts et al., 2013, 2014).

Because of this high spatial and temporal variability it remains challenging to determine changes in SMB of coastal East Antarctica in recent decades. For Dronning Maud Land (DML) alone, conflicting trends have been reported. For example
Altnau et al. (2015) found a negative trend for the last 50 years for coastal DML using a compilation of firn cores, which matches with the results of Medley and Thomas (2019), who, even though their results show a generally positive SMB trend for East Antarctica, also found a negative SMB trend for coastal DML. On the other hand (Philippe et al., 2016) observed a





positive SMB trend in a 120 m ice core in coastal DML for the last 50 years. Similar results were found by Thomas et al. (2017) who report a slightly positive trend for DML in the last 50 years, whereas RACMO2 models no significant trend in the

SMB for DML (Lenaerts et al., 2012b).

In order to understand these seemingly conflicting results, it is important to highlight and further investigate the high spatial SMB variability in DML. One of the main drivers for this spatial SMB variability are the numerous ice rises covering the coast of DML (Lenaerts et al., 2014). Ice rises are elevated local pinning points of the otherwise flat ice shelf, where the ice is grounded on topography and which are surrounded entirely (isles) or mostly (promontories) by the ice shelf (Matsuoka et al.,

2015). Their topography influences the surrounding SMB, by inducing snowfall, especially on the windward side of an ice rise and blowing snow by alternating the wind speed patterns (Lenaerts et al., 2014). In addition, ice rises are an ideal location to drill ice cores as they represent a local ice divide with low ice flow speeds (Matsuoka et al., 2015), which is necessary to recover past accumulation rates from a constant location in time. However, to interpret SMB rates recovered from ice cores, it is necessary to understand the impact of the ice rise on the regional and local SMB. There are two main processes by which ice

rises control the SMB in coastal East Antarctica, which occur on different scales.

Firstly, in areas of steady winds, like DML, orographic uplift of moist air on upwind slopes of the ice rise leads to high snow accumulation on the windward side of the ice rise and a precipitation shadow on the leeward side of the ice rise (Lenaerts et al., 2014). This process occurs over the scale of the ice rise, which are typically 5 - 50 km wide in coastal East Antarctica. Here we will define the scale from 5 - 50 km as the regional scale.

Secondly, local wind erosion and deposition of snow, especially around the peak of the ice rise, play a large role in the SMB distribution on the ice rise (King et al., 2004). This has been observed around local variations of the surface slope on a scale from 1 - 5 km (King et al., 2004; Drews et al., 2015; Mills et al., 2019; Schannwell et al., 2019). Here we will define the scale from 1 - 5 km as the local scale.

However, while both processes have been observed and modelled separately, not much is known about the relation between

the two as well as the importance of the two processes relative to each other. Here we aim to investigate this connection on the example of two ice rises in Dronning Maud Land (Fig. 1), using extensive field measurements as well as the regional atmospheric climate model RACMO2, version 2.3p2 (Lenaerts et al., 2017; van Wessem et al., 2018) and the distributed high resolution snow evolution model SnowModel (Liston and Elder, 2006). The field data were recorded during the 2017/2018 and 2018/2019 Mass2Ant field campaigns. Mass2Ant is an international project that aims to understand surface mass balance

variability in space and time throughout coastal East Antarctica (Dronning Maud Land), with a focus on potential changes since 1850. The data used here includes four ground penetrating radar (GPR) tracks across two ice rises, hereafter referred to as the FKIR and the TIR (Fig. 1), in the vicinity of the Belgian Princess Elisabeth Antarctica (PEA) station. Additionally, the field dataset includes data from two automatic weather stations (AWS), installed on the eastern and on the western flank of the FKIR, as well as an ice core at the center of the FKIR and several firn cores along the GPR profile (Fig. 1). To connect the

different scales we use the GPR data to reconstruct past accumulation rates across the two ice rises.

We combine the SMB reconstructed from the GPR data with the regional SMB distribution modelled by RACMO2 and the local SMB distribution over the ice rise modelled by SnowModel to quantify the magnitude of the local and regional effects





on the SMB, what drives them and how well they are captured within the models. The goal of this study is twofold: (i) reveal mechanisms that control the SMB on and around an ice rise, and (ii) to better understand how SMB rates recovered from ice cores drilled into ice rises, relate to the surrounding ice shelf.

## 2 Methods and Data

### 2.1 Study area

The two ice rises are located in Dronning Maud Land in coastal East Antarctica close to the Belgian PEA station. Both ice rises are, to be specific, ice promontories and are surrounded by the ice shelf from the east, north and west and connected to the grounded ice sheet to the south (Fig. 1). The FKIR is ~ 350 m high and ~ 25 km wide. It represents a local ice divide and neighbours the Roi Baudouin Ice Shelf to the east. On this ice rise, a 20 km east to west GPR profile was recorded during the 2017 Mass2Ant field campaign, using a 400 MHz antenna. Along this profile, five firn cores were recovered to measure near surface density variations in vertical direction. In addition three 10 km GPR profiles were recorded during the 2018 Mass2Ant field campaign across ice rise immediately west of the FKIR, the TIR, which is ~ 360 m high, ~ 50 km wide and represents a triple junction for the ice flow (Fig. 1).

Additionally, an ice core was recovered at the center of the FKIR and two AWSs were installed on the western and on the eastern flank of the FKIR (Fig. 1). The AWSs measured an average southeast wind direction of ~ 132 °, which leaves the eastern flanks of the ice rises as the windward side and the western flanks of the ice rises as the leeward side.

### 2.2 Ice core data

A 208 m ice core (FK17) was drilled on the FKIR (70.53648 S, 24.07036 E), during the 2017/2018 austral summer. We used the intermediate-depth ice core drill (ECLIPSE ice coring drill, Icefield Instruments, Inc.). A ~ 2 m deep trench was excavated prior to drilling to allow the set-up of ECLIPSE ice core drill. To cover this part, a short core (~ 9 m, FK18) was drilled during following field season (2018/2019 austral summer). The cores were cut at a vertical resolution of 50 cm in the field to ease their processing. The ice cores were processed in a freezer laboratory in Université libre de Bruxelles (Brussels, Belgium). The ice cores were cut in sticks with a clean bandsaw. The sticks (~ 3 x 3 x 50 cm) were used for trace ions measurement and the outer part of the stick were used for water stable isotopes measurement. Then the width, length, and weight of the ice core sticks were measured three times and averaged to reconstruct the density profile. Forty-three discrete sticks out of the top 125 m of the ice core were measured to obtain a density profile. Preliminary dating was achieved by counting the seasonal signal of the water stable isotopes ($\delta^{18}$O and $\delta$D). Water stable isotopes were measured using a cavity ring down spectrometer (CRDS) (Picarro, L 2130-i). Samples at 5 cm resolution were melted in a refrigerator prior to analysis. The values are expressed relative to the Vienna Standard Mean Ocean Water (VSMOW) (Philippe et al., 2016; Inoue et al., 2017). Analytical precision for $\delta^{18}$O is < 0.15 and for $\delta$ D is < 0.5 ‰. Note that this preliminary dating from the FK17 ice core needs to be further confirmed by multiple proxies such as trace ions (Na$^+$, Cl$^-$, non-sea salt SO$_4^{2-}$). The annual snow accumulation record was obtained from the



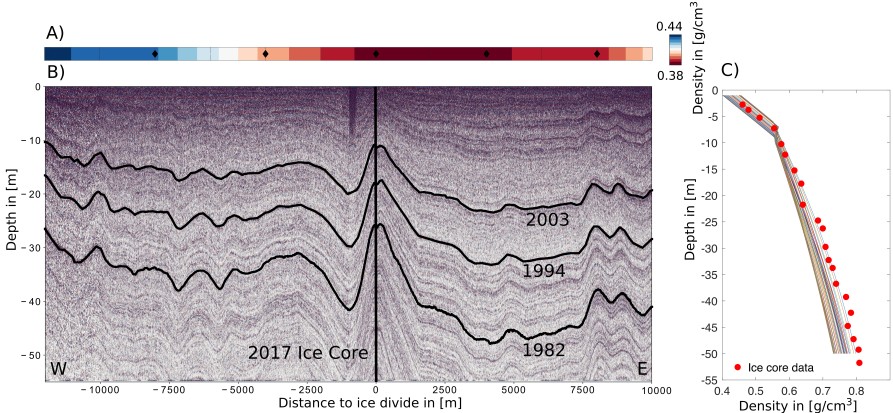

**Figure 2.** (A) interpolated surface density, (B) GPR profile across the FKIR from west to east (red line on the FKIR in Fig. 1), and (C) the modelled density curves together with measured density from the FK17 ice core. The surface density is linearly interpolated from firn core measurements, the black diamonds mark the location of the firn cores. The black lines in (B) show the three tracked horizons together with their dating and the vertical black line shows the location of the FK17 ice core.

thickness of annual layers after ice core dating was done. The thickness of the annual layers was converted into mw.e. using

the density measurements of the ice core.

## 2.3 SMB from Ground penetrating radar

GPR is a technique that uses radar beams emitted into the ground and records the time it takes for the signal to return, after being reflected on internal layers within the snowpack. These internal reflection horizons (IRH) are created by changes in conductivity, density, and ice crystal orientation and represent former surface layers (Fujita et al., 1999; Callens et al., 2016).

The two-way travel time of the signal is then converted into depth by multiplying it with the radar beam velocity within the snowpack and dividing by two. Here we employed a 500 Mhz antenna, dragged behind a skidoo to survey the first 50 m of the snowpack in vertical direction.

To reconstruct the SMB out of the GPR data we used the shallow layer assumption, in which snow accumulation and densification control the IRHs in the upper layers of the snow column while vertical strain plays a larger role in the deeper

layers (Waddington et al., 2007). Following this assumptions one only needs the depth of an IRH, the age of the IRH and the density profile with depth to calculate the average SMB. This is because the depth of the IRH layer tracked through the GPR profile provides information about the volume of snow which accumulated since the IRH formed the surface of the snowpack. Combined with the layer age information this provides the volume of snow accumulated in a certain time frame. If additionally the density distribution with depth of the snowpack is known, it is possible to calculate the snow mass which accumulated since

a certain point in time, with the layer age information from the ice core. This value can be interpreted as the average SMB over the time period, as SMB is a measure of snow mass accumulated per year.



To obtain the depth and age of the IRH's as well as the density distribution with depth, for the profile across the FKIR, we followed a step wise process. The goal of this process is to reconstruct the average SMB across the profile for three time periods within the last 40 years.

1. The fist step was to obtain the density distribution with depth, as this is needed to a) calculate the SMB from the layer depth and b) calculate the increase of radar velocity with depth. In order to do get the density distribution with depth we used the Herron-Langway (Herron and Langway, 1980) firn densification model. The model requires surface density, average surface temperature and average accumulation rates as input. We used the surface density from firn cores along the GPR profile and interpolate that linearly (Fig. 2 A). For the average surface temperature -17 °C was used, which is the average surface
temperature of the last 36 years according to RACMO2. For the average accumulation rates we used an iterative procedure, where we first set the accumulation to 0.55 m w.e./yr across the whole profile, and then used the results of the first run as input for the second. This way we created an ensemble of 220 density curves along the GPR profile, one for every hundred meters, to account for lateral changes in the density distribution with depth. The modelled density curves are in good agreement with the densities measured in the FK17 ice core in the center of the ice rise (Fig. 2 C).

2. The second step was to trace three IRHs and date them using the age information of the 2017 Mass2Ant ice core in the center of the GPR profile (Fig. 2). The deepest IRH represents the surface from 1982, followed by the IRH representing 1994 and 2003. This layer depth was first calculated from the two-way travel time using a constant radar velocity and later updated for the increase of radar velocity with density following the mixing formula of Looyenga (1965). The combination of tracked layers and ice core provide the depth and age information for the SMB.

3. Finally we combined the density model, one layer depth value for every hundred meter along GPR IRHs and the age dating from the ice core to calculate 220 SMB values for each time interval, along the profile (Fig. 3 B). The error bars around the SMB profile represent the uncertainty due to the age measurement and the density model.

For the three profiles across the TIR (Fig. 3) a different approach, without dating the tracked IRHs, was used, as there is not yet an age information for the IRHs available, at the point of writing this study. Due to the lack of dating information,
we worked with the relative spatial SMB distribution across the ice rise, which is independent of the age information. This was accomplished by tracking one continuous IRH across all three adjacent/connected profiles (Fig. 3) and assigning it with the age of its average depth according to the 2017 ice core recovered at the FKIR. To address the potential uncertainty as a result of this relative approach, we introduced an uncertainty of $\pm 5$ years to the age. Additionally, due to a lack of surface density measurements recorded at the TIR profiles, we set the surface density to the average value of 400 $kg/m^3$ for the
Herron-Langway density model. Consequently the SMB profiles two, three and four across the TIR (Fig. 3 D - F) can only be interpreted in terms of their relative SMB variations across the profiles, while disregarding their absolute values. However, since the three profiles on the TIR are connected in a triangle, it was possible to trace the same IRH through all of them and spatially interpolate the SMB within the triangle (Fig. 3 C).





## 2.4 RACMO2

RACMO2 is a state of the art regional atmospheric climate model, which similar to for example MAR (Agosta et al., 2019) and COSMO-CLM (Souverijns et al., 2019) is able to provide accurate SMB simulations in polar regions. RACMO2 was developed by the Royal Netherlands Meteorology Institute (KNMI) and combines the High Resolution Limited Area Model (HIRLAM, Undén et al. (2002)) numerical weather prediction model with the European Centre for Medium-range Weather Forecasts Integrated Forecast System (ISF) physics (ECMWF, 2009; Noël et al., 2015; Lenaerts et al., 2012b)). Here we are

using the latest version RACMO2.3p2 for Antarctica (van Wessem et al., 2018). RACMO2.3p2 includes a multilayer snow model that simulates the effect of heat diffusion, compaction, melt water percolation, retention and refreezing and grain size evolution on the surface temperature and albedo. Furthermore, a bulk snowdrift model is embedded in RACMO2 (Lenaerts et al., 2012a). Here, data from a 5.5 km resolution simulation for East Antarctica covering the time span of 1979 - 2016 are used, of which we used the time period from 2011 to 2016.

## 2.5 SnowModel

SnowModel is a distributed snow evolution model (Liston and Elder, 2006). It consist of four connected modules, which model and spatially interpolate: the meteorological input quantities (MicroMet), the energy balance (EnBal), the evolution of the snowpack (SnowPack) and the redistribution of blowing snow by erosion and deposition (SnowTran-3D). The spatial interpolation of the meteorological input data follows the Barnes objective analysis scheme, which uses a Gaussian weighted

average technique (Koch et al., 1983). SnowPack simulates the changes of snow depth for a, in our case, single snow layer and does not consider any snow micro structure. SnowTran-3D updates this snow depth depending on the amount of saltation, suspension and blowing snow sublimation (Liston and Sturm, 1998). The spatial resolution of SnowModel is limited by the spatial resolution of the DEM. Here we used the TanDEM-X 90 m DEM (Lenaerts et al., 2016).

In terms of meteorological input, SnowModel relies on time series of five quantities: air temperature, relative humidity, wind

speed, wind direction and snowfall for at least one point within the area of interest. Daily RACMO2 data from the grid cells surrounding the locations of the two AWSs was used as meteorological input for SnowModel.

In addition to the DEM and the meteorological variables, SnowModel requires a number of additional input parameters. The most consequential ones are the curvature length scale (distance over which curvature is calculated) of the topography, the weighting between curvature and slope for the wind model, the threshold shear velocity and the roughness length. Regarding

the weighting between curvature and slope, we focused on the effect of the curvature on the wind field, setting the curvature weighting to 1.0 and the slope weighting to 0.0. We did this to focus on processes located at the peak of the ice rise, where the ice core was drilled. At this part of the ice rise the curvature is large, while the slope is zero. For the curvature length scale we used 100 m. Another important input parameter is the roughness length, which together with the threshold shear velocity determines the wind speed necessary to create erosion. The roughness length describes the theoretical height above

the surface, where the wind speed would reach zero (Blumberg and Greeley, 1993). Here we used a roughness length of 0.005 m and 0.6 m/s for the threshold shear velocity. Even though the chosen roughness length is an order of magnitude larger





than typically reported for the Antarctic ice sheet (König-Langlo, 1985; Bintanja and Van Den Broeke, 1995), in wind erosion dominated environments with large sastrugi forming, larger roughness lengths have been found (Amory et al., 2017). Clifton et al. (2006) observed a strong relationship between threshold shear velocity and snow density, where values larger than 0.6 would be expected for the threshold shear velocity for a near surface snow density of above 300 $\mathrm{kg/m^3}$. Combined with the chosen roughness length, a threshold shear velocity of 0.6 $\mathrm{m/s}$ results in 10 $\mathrm{m}$ threshold wind speeds of around 11.4 $\mathrm{m/s}$ to initiate erosion. Based on the frequency with which RACMO2 simulated wind speeds exceed 11.4 $\mathrm{m/s}$ for the ice rise, we estimated that with those settings, we have an appropriate representation of erosion frequency in SnowModel.

## 3 Results

### 3.1 Regional surface mass balance around the ice rises

To investigate the regional SMB around the ice rises we utilize the SMB modelled by RACMO2 (Fig. 4) as well as the SMB reconstructed by the GPR data (Fig. 3). Both SMB datasets qualitatively agree on the SMB variability over the ice rises, but differ on the amount and contrasts between windward and lee side. For our region of interest in Dronning Maud Land (the whole extent of Fig. 4 A), RACMO2 models an average SMB of ~ 0.35 mw.e./yr across the ice rises and ice shelf. However, along the windward flanks of the FKIR, the TIR and the DIR the modelled SMB values are substantially higher around ~ 1.0 mw.e./yr (Fig. 4). These high SMB values on the windward flanks of the ice rises are accompanied with low SMB values on the leeward flanks of the ice rises around ~ 0.2 mw.e./yr. We can observe a similar SMB difference across the FKIR in the SMB reconstructed from the GPR data. The GPR profile across the FKIR (Fig. 3) shows SMB values between 0.4 and 1.0 mw.e./yr, with an average SMB across the ice rise of ~ 0.7 mw.e./yr in the period of 1982 to 1994 and ~ 0.6 mw.e./yr between 1994 and 2003 as well as ~ 0.65 mw.e./yr in the most recent time period of 2003 to 2017. The SMB values for all three time periods are consistently higher on the windward side of the ice rise than on leeward side of the ice rise (Fig. 3 B). This windward-leeward difference is largest between 2003 and 2017 with ~ 0.24 mw.e./yr average SMB difference and lowest in the period of 1994 and 2003 with ~ 0.07 mw.e./yr. However, while RACMO2 models a ~ 5 times higher SMB on the windward side of the FKIR than on the leeward side, the GPR profile 1 across the FKIR measures at most a ~ 1.5 times larger SMB on the windward side of the ice rise than on the leeward side. That implies that for FKIR, the difference in SMB between windward and leeward side is considerably larger in RACMO2 than in the observations (Fig. 5). This is in agreement with Agosta et al. (2019), who found that RACMO2 indeed overestimates snowfall on the windward site of topography. However, the average SMB across the ice rise (parallel to profile 1 of figure 3) in RACMO2 is similar to the SMB from the GPR for the observed time period of 1984 to 2017, with ~ 0.54 mw.e./yr and ~ 0.65 mw.e./yr respectively.

For the TIR RACMO2 models between ~ 2 and ~ 2.40 times higher SMB values on the windward side of the ice rise than on the leeward. This difference is stronger in the north of the ice rise where SMB values are around ~ 0.6 mw.e./yr and lower in the south where SMB values are around ~ 0.5 mw.e./yr (Fig. 4). Here the SMB reconstructed from the GPR data shows a similar difference of ~ 2 times higher SMB values on the windward side than on the leeward side, but in contrast to RACMO2

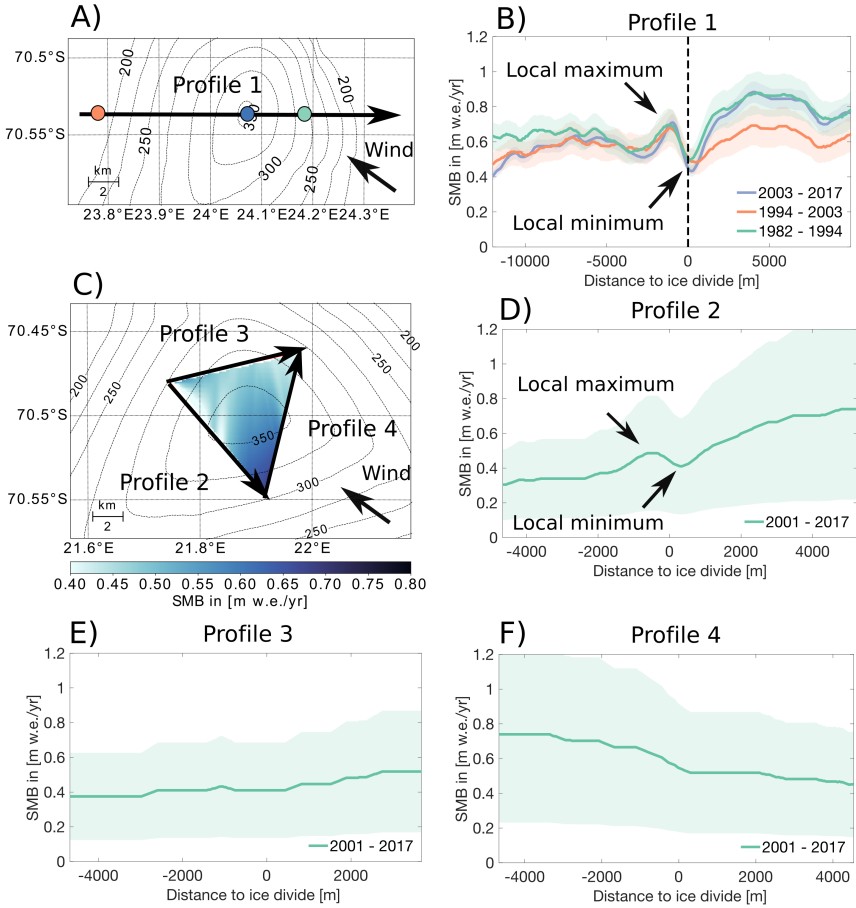

**Figure 3.** Topography of the FKIR, the location of the two AWS's (green and orange dots), the drilling site of the FK17 ice core (blue dot) and the GPR tracks recorded during the 2017 Mass2Ant field campaign (A). Also shown is the SMB reconstructed from the GPR profile on the FKIR (B) as well as the topography and the GPR tracks on the TIR (C), where three SMB profiles were reconstructed from GPR profiles for a single time window (2001-2007), by tracking a single IRH of age deduced from the depth-age relationship at the FKIR ice core (D,E,F). The topography of the ice rises shown in (A) and (C) is based on the 90m TanDEM-X digital elevation model (© DLR 2018).

(Fig. 4 A) the highest values are found in the south east of the ice rise instead of the north east (Fig. 3 C). Now one would
intuitively expect SMB to be highest in the south east sector of the ice rise, where rising air provides enhanced precipitation.

In general, the regional difference in SMB between the windward and the leeward side of the FKIR seems to be largely driven by a difference in snowfall. We can observe this in the distribution of the modelled SMB components across the ice rise (Fig. 5 A). In figure 5 A we see that the SMB curve across the ice rise largely follows the snowfall curve. Sublimation and wind redistribution also play a role for the SMB but sublimation is relatively constant in space and does not seem to contribute to the
SMB difference between the windward and the leeward side of the ice rise. Wind erosion also seems to only play a negligible role in creating the large scale SMB difference between the windward and the leeward side of the ice rise (Fig. 5), as RACMO2

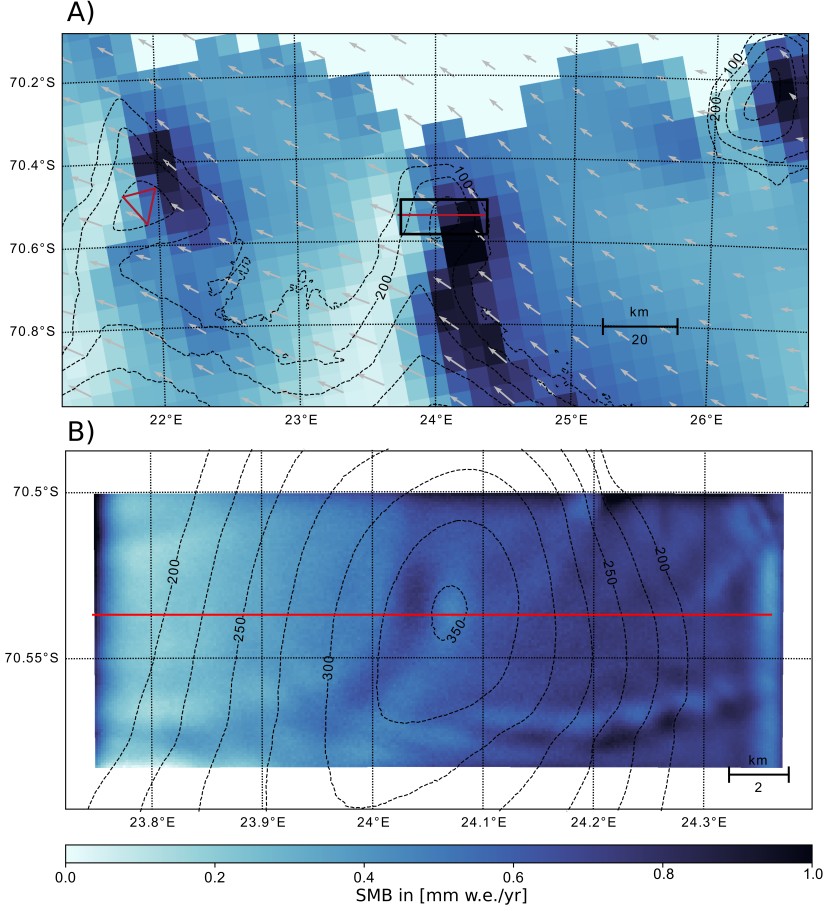

**Figure 4.** Overlays of the SMB modelled by RACMO2 (A) and SnowModel (B) for the time period of 2011 to 2017 with topography contour lines based on the 90m TanDEM-X digital elevation model (© DLR 2018) and RACMO2 wind vectors (grey arrows). The simulated area by SnowModel is marked in (A) with a black box. The red lines show the location of the recorded GPR tracks.

models relatively constant erosion across the whole ice rise. In RACMO2, this eroded snow is then deposited downwind of the ice rise on the surrounding ice shelf (Lenaerts et al., 2014). However, with sublimation and wind redistribution being relatively constant across the FKIR, the difference in snowfall seems to be the controlling factor for the difference in SMB. This confirms the narrative that the large scale difference in SMB is a result of orographic precipitation on the windward side of the ice rise (Lenaerts et al., 2014).

### 3.2 Local surface mass balance at the ice divide

Local minima in SMB can be identified in the GPR data at the ice divide (the highest point of the ice rise) as well as local SMB maxima within the first 2000 m west of the ice divide in the SMB profile across the FKIR (Fig. 3 B). These large amplitudes





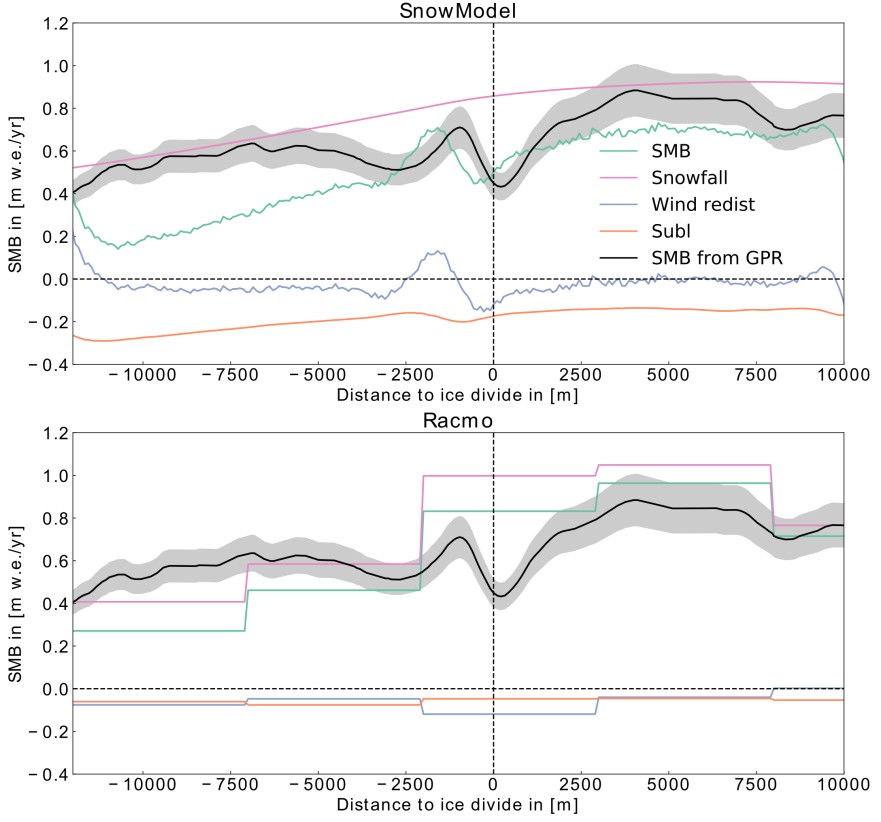

**Figure 5.** Profiles showing the total SMB and the different components of the SMB across the west to east profile of the FKIR modelled by (A) SnowModel and (B) RACMO2. SMB reconstructed from the GPR data is also shown together with an estimated error in light grey. The location of the profile is shown in Fig. 4.

in the GPR layers may be the result of accumulation differences and therefore represent local SMB differences, but they may also be the result of internal ice deformation resulting from higher vertical velocities at the flow divide compared to the flanks, the so called Raymond effect or Raymond bump (Raymond, 1983). To test whether or not this anomaly around the peak of the ice rise is an accumulation feature controlled by the SMB or the result of the Raymond effect we used a technique described by Vaughan et al. (1999). If the amplitudes of the arches in the GPR data increase linear in size with depth it is the result of

snow accumulation, whereas if the amplitudes increase quadratic it is the result of internal deformation (Raymond effect). As shown in Fig. 6, the amplitudes in our data increase linear in size with depth for profile one and are therefore interpreted as a result of accumulation. This argument is supported by the results of Drews et al. (2015), who applied the same method at the neighbouring DIR and also came to the conclusion that the shallow arches found in their GPR data are a result of accumulation differences and not internal deformation.

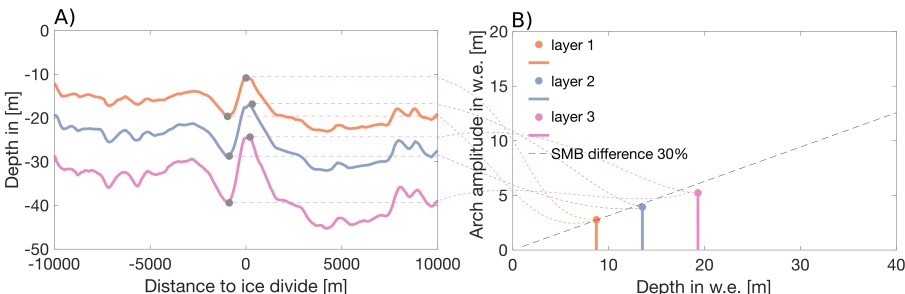

**Figure 6.** A) shows the depth of the tracked IRH from the GPR data (Fig. 3) across the FKIR. B) shows the arch amplitudes around the ice divide (in w.e.) versus the depth of the arches. The dashed line shows the theoretical size of the arch amplitudes with depth for an SMB difference of 30 %.

This anomaly of low SMB at the divide and high SMB west of it is consistent through all three time periods observed at the FKIR and has an average amplitude of ~ 0.22 mw.e./yr. The SMB values at the ice divide are ~ 0.45 mw.e./yr during the most recent time period of 2003 to 2017, ~ 0.51 mw.e./yr from 1994 to 2003 and ~ 0.50 mw.e./yr from 1982 to 1994. These values agree with the average accumulation rates for the same time periods determined from the ice core at the center of the ice rise of ~ 0.40 mw.e./yr, ~ 0.50 mw.e./yr and ~ 0.48 mw.e./yr, respectively (Fig. 7). We also observe a similar SMB minimum at

the ice divide in profile 2 across the TIR as well as a local SMB maximum within 2000 m down wind of it. However, there the SMB minimum is ~ 500 m south east of the ice divide (Fig. 3 D). The SMB varies by ~ 0.1 mw.e./yr between the minimum at the divide and the down wind maximum, which is substantially smaller than at the FKIR. In addition to this, the anomaly is only present in profile 2 of the three profiles across the TIR (profiles 2,3,4 Fig. 3), even though all profiles cross a local ice divide.

To investigate the role of snow redistribution by wind, we use SnowModel with RACMO2 forcing for the area on the FKIR. SnowModel reproduces the anomaly of low SMB on the peak of the ice rise and a local maximum west of the peak (Fig. 5). Furthermore, SnowModel provides the individual SMB components, which reveal that this anomaly in the SMB is a result of high negative wind redistribution (erosion) at the peak and high positive wind redistribution (deposition) within the next 2 km west of the peak. This indicates that snow is eroded at the top of the ice rise where wind speeds are high and deposited within

2 km leeward of the ice divide. This is in agreement with the observations made by King et al. (2004) who observed similar local SMB variations across the Lyddan ice rise in Antarctica and explained them as a result of snow redistribution by wind. The absence of the local minima at the ice divide on profile 3 and 4 across the TIR (Fig. 3) provides further evidence that this is a wind driven accumulation feature and not a result of internal deformation. It seems that positive curvature of the topography, parallel to the dominant wind direction, is necessary to create this erosion and deposition feature.





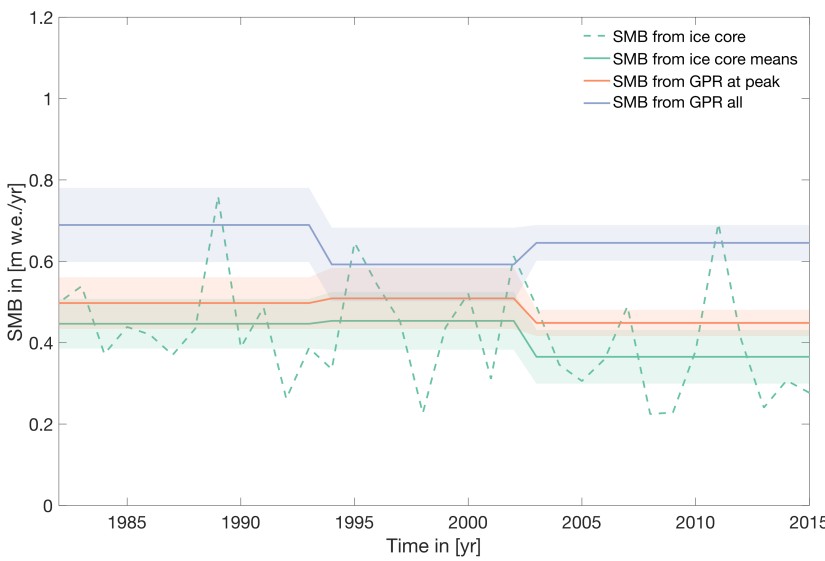

**Figure 7.** Shows the SMB in time recovered from the FK17 ice core (dashed green line), as well as the SMB recovered from the FK17 ice core averaged for the 3 time periods 1982 - 1994, 1994 - 2003, 2003 - 2015 (solid green line). The solid orange line is the SMB for the same three time periods reconstructed from GPR profile 1 at the peak of the FKIR, wheras the solid blue line is the average SMB for the three time periods reconstructed across GPR profile 1.

## 4  Discussion

Overall the SMB of an ice rise seems to be controlled by two different mechanisms, a larger scale systematic difference of higher snowfall at the windward side of an ice rise, which affects the SMB of the ice rise as a whole, and a localized effect of wind erosion and deposition around the peak of the ice rise. This is in agreement with Drews et al. (2015), who observed similar effects on the neighboring DIR explaining the large scale difference as a result of higher snowfall on the windward side and suggesting wind redistribution as a driving force of the small scale variations. However, they mention that this might not be a generic ice rise feature as Vaughan et al. (1999) and Conway (1999) did not find any evidence for wind erosion at Fletcher Promontory and Roosevelt Island respectively. However, those studies used low frequency radar with high penetration depth to investigate internal structures within the ice, which consequently had low resolution near the surface, such that it is possible that the effect of wind erosion on the peak of the ice rise was not resolved. Since we found these effects to occur on both ice rises, congruent with studies from other locations (Derwael and Lyddan ice rise (Drews et al., 2015; King et al., 2004)), we conclude that these features are likely generic for ice rise features in at least East Antarctica, possibly all of coastal Antarctica.

To investigate the implications of these two processes for ice rises as ice core drilling locations, we can compare the SMB recovered from FK17 to the SMB from GPR across the FKIR and to the SMB form the GPR at the location of FK17. The ice core at the center of the FKIR yields an average SMB of ~ 0.45 $\mathrm{mw.e./yr}$ between 1982 and 2015. This is in agreement with



275 the average SMB reconstructed from the GPR at the ice core location of ~ 0.48 mw.e./yr for the same time period (Fig. 7). However, as we showed earlier, the center of the ice rise is affected by substantial erosion. This results in the SMB being lower at the ice core location than the average SMB of the ice rise across profile 1 for the same time period, which is ~ 0.65 mw.e./yr. However, the average SMB along profile 1 of the FKIR is substantially larger than the general SMB of the surrounding ice shelf due to the high amount of snowfall on the windward flank of the ice rise. Even the average SMB values on the leeward

280 side of the FKIR of ~ 0.56 mw.e./yr are still well above what RACMO2 models for the average SMB of the surrounding ice shelf area with ~ 0.35 mw.e./yr. So unlike in RACMO2 where SMB values are as low as ~ 0.27 mw.e./yr on the leeward side of the ice rise, we do not observe a strong precipitation shadow on the leeward side of the FKIR in the GPR data. This means that the peak of the ice rise represents the location within GPR profile 1 where the reconstructed SMB is closest to what one would expect for the surrounding ice shelf. Therefore, in case of the FKIR, it seems like the erosion at the ice divide partially

285 cancels out the higher SMB values due to orographic uplift and results in an overall lower SMB at the ice core location, which better resembles the surrounding shelf.

 However, the observed difference between the local SMB minimum at the peak of the ice rise, where the snow gets eroded, and the SMB maximum just upwind of the peak, where the snow gets deposited, is very large at the FKIR (~ 0.22 mw.e./yr) compared to the neighboring TIR and DIR. At both of these locations this erosion and deposition feature has an amplitude of

290 ~ 0.1 mw.e./yr. Surprisingly this is the case despite the generally lower average wind speeds of ~ 8.8 m/s across the FKIR, compared to ~ 9.6 m/s across the TIR (according to RACMO2). Also at the FKIR the erosion and deposition feature was largest in the time period from 2003 to 2017, where the average wind speed was slightly lower than from 1982 to 1994 and 1994 to 2003, according to RACMO2. Therefore it seems that wind speed is not the only determining factor for the local erosion feature.

295 Theoretically erosion occurs if the local wind speed exceeds a certain wind speed threshold, which in itself depends on the surface roughness length and the threshold shear velocity (Clifton et al., 2006). In addition it seems that threshold shear velocity is largely dependent on near surface snow density (Clifton et al., 2006). Therefore a possible explanation for the higher erosion at the FKIR could be that the magnitude of the erosion feature depends on the overall snowfall, where higher snowfall would create larger erosion by enhancing the availability of fresh low density snow, which in turn is easier to erode, as it has a

300 lower threshold shear velocity. However, in SnowModel the threshold shear velocity does not change with snowfall and more advanced models (e.g. Alpine3D (Lehning et al., 2006)), which include snow micro structure, would be necessary to model the effect of a changing threshold shear velocity with snowfall. In fact, there is observational evidence for an increase of erosion with snowfall. For example a recent study by Souverijns et al. (2018), who were using a set of remote sensing instruments at a study cite close to PEA station, found that snowfall events only led to accumulation 60% of the time, while leading to ablation

305 40% of the time due to the erosion of the freshly fallen snow and confirmed earlier studies in that the availability of fresh snow is more important for erosion to occur than high wind speeds (Gallée et al., 2001).

 Another indicator for the increase of erosion with snowfall is that, the SMB variability on the windward side of the FKIR between the three observed time periods is almost three times higher (~ 0.17 mw.e./yr at maximum) than the temporal SMB variability at the peak of the ice rise (~ 0.06 mw.e./yr at maximum). This would imply that locally, around the peak of the ice



rise, the regional processes of high SMB on the windward side, due to higher snowfall, would be partially canceled out by the erosion process. A consequence of this is, that the absolute accumulation values measured from the ice core at the peak of the ice rise resemble more closely the SMB values expected for the surrounding ice shelf, but also that they show less sensitivity to the temporal snowfall variability on the ice rise itself. For example the overall SMB across the GPR profile is higher in the time period from 2003 to 2017 than from 1994 to 2003. But the SMB reconstructed from the GPR at the peak of the ice rise as well as from the ice core shows a lower SMB in the time period from 2003 to 2017 than from 1994 to 2003 (7). However, for now we have only observed this at the FKIR, since we do not have a temporal resolution for the TIR data to test this in more detail. More observations, at other ice rises, where GPR data and ice core data are combined would be needed to confirm this hypothesis. A possible candidate for this would be the neighboring DIR where all these data sets have been collected. In any case our results highlight the value of GPR data around an ice core to put SMB values recovered from the ice core into context.

## 5 Conclusions

Here we combine in situ GPR measurements and ice core data with state of the art modelling results to investigate the spatial variations across two ice rises in Dronning Maud Land as well as the spatial and temporal variations across one of the ice rises for the time period of 1982 to 2017. We identify two main processes which influence the SMB across the ice rises, a regional snowfall driven process of higher SMB on the windward side of the ice rises due to orographic uplift as well as a local wind driven erosion process, where snow is eroded at the peak of the ice rise and deposited within a couple of kilometers down wind of the peak. At the FKIR, where we have both GPR data and an ice core available, the SMB reconstructed from the GPR shows that both processes play a role for the SMB at the drilling location. For this ice rise the erosion at peak of the ice rise locally compensates for ice rise wide temporal variations in snowfall. As a result of this, the absolute SMB values from the ice core are closer to the average SMB values of the surrounding ice shelf as they are less affected by the high SMB values created by orographic uplift on the windward side of the ice rise. On the other hand however, this means that they fail to capture the temporal changes in snowfall which are present on the windward side of the ice rise.

*Data availability.* Data available on request from t.kausch@tudelft.nl and will be uploaded to a public repository with DOI on eventual final publication

*Author contributions.* TK,SL,JTML,JLT and NW conceived this study. TK led the writing of the manuscript, and performed the analysis of the data. FP, JLT, MI, SW, NW, SL, JTML and SS gathered the field data. WJB and JTML contributed to the development of RACMO2.3 and provided the dataset. JLT, MI and SW analysed the ice core and MI and SW wrote the respective method chapter about the ice core. All authors contributed to discussions on writing the manuscript.



*Competing interests.* The authors declare that they have no conflict of interest.

*Acknowledgements.* This work has benefited from BELSPO Research Contract, grant BR/165/A2:Mass2Ant. Thore Kausch and Stef Lher-
340  mitte were supported by the NWO Polar Program, grant ALWPT.2016.4. J. T. M. Lenaerts acknowledges support from the National Aeronat-
ics and Space Administration (NASA), grant 80NSSC18K0201. Sarah Wauthy is a reasearch fellow of F.R.S.-FNRS. In addition, we would
like to thank the German Aerospace Center (DLR) for providing the 90 m tanDEM-X DEM and Glen E. Liston for developing and providing
SnowModel. Finally we thank the international Polar Foundation (IPF) for the logistic support during the Mass2Ant field campaigns.



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
