# Peer review of "Impact of coastal East Antarctic ice rises on surface mass balance: insights from observations and modeling"

_The Cryosphere, 2020_

## Referee Comment (RC1) · Anonymous Referee #1 · 17 Apr 2020

**Review of "Impact of coastal East Antarctic ice rises on surface mass balance:**

**insights from observations and modelling", by Kausch et al. (TC-2020-66)**

**General comments**

In this well-written paper, the authors use a comprehensive set of measurements of surface mass balance (SMB) across two Antarctic ice rises together with meteorological observations, a regional atmospheric model and a SMB model to investigate the factors that control the spatiotemporal variation of SMB across these features. The authors conclude that variations in SMB across the ice rises is controlled by both orographic forcing of snowfall and wind-borne transport of snow, with the two effects partially cancelling each other. As a result, SMB at the ice rise summit is close to its value on the surrounding ice shelf. The results have important implications for the interpretation of ice core records of SMB obtained from ice rise summits.

The paper is very clearly written and the methodology is sound and is well-explained. The conclusions are soundly based on the data and the analysis. I recommend publication of the paper following minor revisions, as detailed below.

**Main points**

1. Section 2.5. It is not entirely clear to me whether the values used for the various SnowModel parameters were chosen *a priori* or were used to "tune" the model to obtain the best fit to the observations. Tuning is a perfectly valid approach, but you should clearly state if that is what was done. Did you carry out any validation of the SnowModel wind field against your two AWSs – very little use appears to have been made of this source of data? You justify using a rather large (5 mm) value for roughness length on the grounds that Amory and others have measured similar values over the large sastrugi fields found in a strong katabatic wind regime in Adélie Land. Did you actually observe similar large sastrugi on your ice rises? I'm not sure how you can be certain that your parameter choices give you an "appropriate representation of erosion frequency" (lines 187-188) unless you have observations to validate this.

2. Lines 284-286: "Therefore, in case of the FKIR, it seems like the erosion at the ice divide partially cancels out the higher SMB values due to orographic uplift and results in an overall lower SMB at the ice core location, which better resembles the surrounding shelf." Is this likely to be a universal result for ice rises? Orographic enhancement of precipitation on the upwind side probably scales with the broad-scale topographic slope on this side, while erosion at the ice divide will scale with curvature at the summit. Is it just coincidence that the two are approximately equal at FKIR or do the fundamentals of ice dynamics mean that this ratio should be the same for ice rises of any scale?

**Minor points and typographical corrections**

1. Line 27: "interannual" (one word)
2. Line 40: Insert a comma after "ice rise"
3. Line 62: Maybe spell out the full names of the TIR and FKIR when you first mention them?
4. Line 91: "sticks" (plural)
5. Figure 2: The black diamonds are not that easy to see. Maybe use markers above the colour bar instead?
6. Line 154: IFS , not ISF

7. Line 170-171: What does the RACMO2 orography look like when compared to TanDEM-X?
8. Section 3.1: There are a lot of numbers given in the text. It might be useful to summarise them in a table or bar chart.
9. Line 234: "linearly"
10. Line 288: "downwind" instead of "upwind"?

---

## Referee Comment (RC2) · Anonymous Referee #2 · 27 Apr 2020

**1   General**

This paper presents data and model results for surface mass balance in the region of two East Antarctic ice rises. The authors show using ground-penetrating-radar-based estimates of SMB and results from RACMO2 and SnowModel that SMB varies on a local scale, due to erosion and redeposition by wind, and on a regional scale, due to orographic lifting. Neither of these results are surprising, but the paper argues the implications for ice-core records recovered on ice rises.

[Figure]

**2 Main issues**

- In section 2.5 a number of parameters are described about the SnowModel, and it's not that clear if the 'knob-twiddling' was to match observations or actually based on a-priori information and assumptions. It would be good to make this more clear.

- It is not that clear how much the analysis from the second ice rise, TIR, really adds to the result. The SMB estimates from TIR can only be considered in a relative sense, and the GPR transects from TIR do not really capture the features that are prominently discussed for the FKIR.

- The implications of the work should be made more prominent. This work has conclusions that impact the interpretation of ice core records from ice rises, but these implications are only given a few sentences near the end of the discussion, and only just mentioned in the abstract and conclusions.

**3 Small, line-by-line stuff:**

- Figure 1: wind-rose plot is difficult to read at magnifications under 200%- suggest reworking that part of the plot if it's important, or eliminating it and replacing with a 'prevalent wind derection' arrow. The black dot at the summit of FKIR is not called out explicitly- presumably it is the ice core location but not clear in the caption- add it in.

- Line 27: hyphenate "inter-annual"

- Line 52, set of references: this actually goes back to Black and Budd 1964 (who King et al cite)- might be good to cite them too!

Black, H. P., and W. Budd (1964), Accumulation in the region of Wilkes,Wilkes Land, Antarctica,J. Glaciol.,5(37), 3–15.

- Line 62: Might as well name the ice rises before shortening them to abbreviations (I'm assuming the abbreviations are for "T" ice rise and "FK" ice rise).

- Line 70: delete comma after "rises"

- Lines 76, 77: including make and model of the (probaby commercial) GPR would be of interest to GPR folks

- Figure 2: Fonts too small to read even at 135% zoom. enlarge. Also, the IRHs are challenging to see without magnification of close to 300%. Can the contrast be improved, or a different color scheme be employed?

- Line 102: Best not to start a sentence with an acronym, even if it's a common one like GPR.

- Line 104: Niether of these references establishes that the IRH is an isorchron, and you really should include one (Callens in particular uses the vague "generally accepted" language that should be avoided. A good early reference for this is Spikes et. al., 2004 (see figure 2):

  Spikes, Vandy B., et al. "Variability in accumulation rates from GPR profiling on the West Antarctic plateau." Annals of Glaciology 39 (2004): 238-244.

- Line 106: In section 2.1 you said you used 400 Mhz. Close enough that the science doesn't change, but choose one (preferably the accurate one) and be consistent (unless of course you used both!).

- Line 106: to clarify the language, suggest changing "the first 50 m of the snow-pack in the vertical direction" to "the shallowest 50 m of the snowpack"
- Line 121: delete "do" from the phrase "in order to do get the density"

- Line 133: Not clear if the initial layer depth mentioned here is part of an iterative procedure or simply part of an initial data analysis step. If the former, describe more fully, if the latter, delete as it's not necessary!

- Line 135: with such detail in the procedure the omission of an actual equation describing mathematically how SMB is derived seems glaring. Suggest adding here.

- Lines 138-148: This approach is fine, and using relative magnitudes of SMB is ok, but I don't think it's appropriate in that case to use absolute values on the y axes of Figures 3 D-F. instead, use some scaled value and leave units off. On line 146 you state that the absolute values should be disregarded- thus leaving them on the figure almost invites misuse of the result.

- Line 161: "it consist of" -> "it consists of"

- Line 165: "for a, in our case, single. . . " awkward. Rewrite.

- Line 191: mention Figure 3 first, and then Figure 4.

- Figure 3: Again, because the SMB estimates in panels D-F are only relative, leave units off the y-axis.

- Figure 4: make the box in panel A (denoting the location of panel B) more prominent.

- Lines 203-204: start referring to figure 5 here; will be useful.

- Figure 5: Needs A and B labels. Also, either repeat legend in panel B (there's plenty of room), or move it outside both panels. Finally, since SMB in each model is what is being directly compared to the GPR SMB, make that curve bold.

- Figure 6: If a figure needs to be removed to save space, this one is a good candidate.

- Lines 238 and 265: I don't see the ice rise DIR actually named anywhere.

- Line 327: "For this ice rise the erosion at the peak…" This sentence is unclear- in particular the expression "ice rise wide temporal variations". I understand the idea but it is not well expressed here- suggest split to a few sentences…

---

## Author Comment (AC1) · 23 Jun 2020

**TC Response to referee 1**

[Figure]

Dear referee,

First of all thank you for carefully reading our manuscript and for your positive review and constructive criticism. The points you are raising are all very much justified and we tried our best to address them as good as possible.

Below you can find your original comments with our answers in blue as well as updated sections from the text in *italic*.

Yours sincerly,
Thore Kausch and co-authors

**1 Referee 1 comments:**

Review of "Impact of coastal East Antarctic ice rises on surface mass balance: insights from observations and modelling", by Kausch et al. (TC-2020-66)

**1.1 General comments**

In this well-written paper, the authors use a comprehensive set of measurements of surface mass balance (SMB) across two Antarctic ice rises together with meteorological observations, a regional atmospheric model and a SMB model to investigate the factors that control the spatiotemporal variation of SMB across these features. The authors conclude that variations in SMB across the ice rises is controlled by both orographic forcing of snowfall and wind-borne transport of snow, with the two effects partially cancelling each other. As a result, SMB at the ice rise summit is close to its value on the surrounding ice shelf. The results have important implications for the interpretation of ice core records of SMB obtained from ice rise summits. The paper is very clearly written and the methodology is sound and is well-explained. The conclusions are soundly based on the data and the analysis. I recommend publication of the paper following minor revisions, as detailed below.

**1.2 Main points**

1. Section 2.5. It is not entirely clear to me whether the values used for the various SnowModel parameters were chosen a priori or were used to "tune" the model to obtain the best fit to the observations. Tuning is a perfectly valid approach, but you should clearly state if that is what was done.

For the roughness length and the threshold shear velocity we indeed used forward modelling to try several different values and in the end chose the ones, which would fit the observations best. This was not clear in the text before. We added a few sentence to clarify this in the text.

*L182-184: For both, the roughness length and the threshold shear velocity we tried different values using forward modelling. We varied the roughness length between 0.05 and 0.00005 m and the threshold shear velocity between 0.3 and 0.8 m/s, and chose the values which qualitatively fit the reconstructed SMB values from the GPR best. In the end we chose a roughness length of 0.005 m and 0.6 m/s for the threshold shear velocity.*

Did you carry out any validation of the SnowModel wind field against your two AWSs – very little use appears to have been made of this source of data?

Regarding the AWS, the reason that we made so little use out of the data is that we only had four months of continuous data available at the time. However we did use the AWS data to justify the "appropriate representation of erosion frequency" but failed to mention this in the text. (There are more details below in the part about the erosion frequency)

You justify using a rather large (5 mm) value for roughness length on the grounds that

Amory and others have measured similar values over the large sastrugi fields found in a strong katabatic wind regime in Adélie Land. Did you actually observe similar large sastrugi on your ice rises?

We indeed observed large sastrugi in the field.

I'm not sure how you can be certain that your parameter choices give you an "appropriate representation of erosion frequency" (lines 187-188) unless you have observations to validate this.

We used the AWS regarding the "appropriate representation of erosion frequency" but failed to mention this in the text. We added a few sentences to clarify this.

*L188-193: Based on the frequency with which RACMO2 simulated wind speeds exceed 11.4 m/s for the ice rise and the frequency with which wind speeds observed by the AWS exceed 11.4 m/s, we estimated that with those settings, we have an appropriate representation of erosion frequency in SnowModel. Between January and April 2018 the AWS on the windward side measured daily wind speeds exceeding 11.4 m/s 50 % of the days. This is in agreement with observations from (Amory, 2020) who observe similar erosion frequencies between January and April for a cite in coastal Adélie Land with a comparable elevation of 450 m.*

2. Lines 284-286: "Therefore, in case of the FKIR, it seems like the erosion at the ice divide partially cancels out the higher SMB values due to orographic uplift and results in an overall lower SMB at the ice core location, which better resembles the surrounding shelf." Is this likely to be a universal result for ice rises? Orographic enhancement of precipitation on the upwind side probably scales with the broad-scale topographic slope on this side, while erosion at the ice divide will scale with curvature at the summit. Is it just coincidence that the two are approximately equal at FKIR or do the fundamentals of ice dynamics mean that this ratio should be the same for ice rises of any scale?

This is certainly an important question to ask. While both processes are always oppositional, it is likely coincidental that the two are approximately equal. This is because both, the erosion at the peak of the ice rise and the orographic enhancement of precipitation do not only depend on the curvature and the slope of the ice rise, but also on a large number of other factors like wind speed, atmospheric moisture content and the height of the ice rise. It would be possible to imagine scenarios in which one process happens in the absence of the other, for example a very dry and extremely windy day, where erosion would occur, while there is no enhanced precipitation because there would simple be none. However, a more sophisticated model, which for example includes the possibility to locally decrease the threshold shear velocity with snowfall, in combination with observations from multiple ice rises would probably be needed to say this for sure. We added a few lines in the discussion to address this.

*L293-299:Therefore, in case of the FKIR, it seems like the erosion at the ice divide partially cancels out the higher SMB values due to orographic uplift and results in an overall lower SMB at the ice core location, which better resembles the surrounding shelf. This is likely a generic feature for ice rises of all shapes as the orographic enhanced precipitation scales with the slope of the ice rise and the erosion at the peak of the ice rise scales with the curvature of the ice rise and both processes are generally opposed to each other. However, the magnitude of each process individually might vary strongly between ice rises as both processes also scale with other factors like wind speed and atmospheric moisture content, in addition to the topography.*

**1.3  Minor points and typographical corrections**

1. Line 27: "interannual" (one word)
Done.
2. Line 40: Insert a comma after "ice rise"
Done.
3. Line 62: Maybe spell out the full names of the TIR and FKIR when you first mention them?
FKIR and TIR were project internal names, named after leading scientist in the Mass2Ant field expedition to the ice rises. However since there are official norwegian

names for both ice rises we decided to rename them. Therefore we changed FKIR to Lokeryggen ice rise (LIR) and TIR to Hammarryggen ice rise (HIR). But we kept the name Derwael ice rise (DIR) since it was already named like this in earlier publications we are citing. We added a note though mentioning the norwegian name of DIR (Kilekollen).

4. Line 91: "sticks" (plural)

Done.

5. Figure 2: The black diamonds are not that easy to see. Maybe use markers above the colour bar instead?

Done

6. Line 154: IFS , not ISF

Done.

7. Line 170-171: What does the RACMO2 orography look like when compared to TanDEM-X?

RACMO uses Bamber et al.'s 2009 1km resolution DEM, whereas TanDEM-X has a 90m resolution for our study area, which naturally makes it significantly more detailed. While the overall topography of both data sets is comparable, the reduced resolution of Bamber et al.'s 2009 DEM compared to TanDEM-X can result in difference of up to 50 - 100 m on the flanks of the ice rises.

8. Section 3.1: There are a lot of numbers given in the text. It might be useful to summarise them in a table or bar chart.

It is true that there are a lot of numbers given in the text, however they are also already displayed in figures 3 and 4, which I personally think should make the numbers easier to grasp than in a table.

9. Line 234: "linearly"

Done.

10. Line 288: "downwind" instead of "upwind"?

Changed to downwind.

[Figure]

[Figure]

**Fig. 1.**

[Figure]

**Fig. 2.**

[Figure]

[Figure]

**Fig. 3.**

A)

B)

SMB in [mm w.e./yr]

**Fig. 4.**

[Figure]

A) SnowModel

B) Racmo

**Fig. 5.**

[Figure]

[Figure]

**Fig. 6.**

---

## Author Comment (AC2) · 23 Jun 2020

**TC Response to referee 2**

Dear referee,
First of all thank you for taking your time read through our manuscript as well as for your positive review and constructive criticism. The points you are raising are all very much justified and we tried our best to address them as good as possible.

Below you can find your original comments with our answers in blue as well as updated sections from the text in *italic*.

Yours sincerly,
Thore Kausch and co-authors

**1  Referee 2 comments:**
This paper presents data and model results for surface mass balance in the region of two East Antarctic ice rises. The authors show using ground-penetrating-radar-based estimates of SMB and results from RACMO2 and SnowMod el that SMB varies on a local scale, due to erosion and redeposition by wind, and on a regional scale, due to orographic lifting. Neither of these results are surprising, but the paper argues the implications for ice-core records recovered on ice rises.

**1.2 Main issues**

In section 2.5 a number of parameters are described about the SnowModel, and it's not that clear if the 'knob-twiddling' was to match observations or actually based on a-priori information and assumptions. It would be good to make this more clear.

For the roughness length and the threshold shear velocity we indeed used forward modelling to try several different values and in the end chose the ones, which would fit the observations best. This was not clear in the text before. We added a few sentence to clarify this in the text.

*L182-184: For both, the roughness length and the threshold shear velocity we tried different values using forward modelling. We varied the roughness length between 0.05 and 0.00005 m and the threshold shear velocity between 0.3 and 0.8 m/s, and chose the values which qualitatively fit the reconstructed SMB values from the GPR best. In the end we chose a roughness length of 0.005 m and 0.6 m/s for the threshold shear velocity.*

It is not that clear how much the analysis from the second ice rise, TIR, really adds to the result. The SMB estimates from TIR can only be considered in a relative sense, and the GPR transects from TIR do not really capture the features that are prominently discussed for the FKIR.

TIR mostly adds the information that the features we see are wind driven SMB features not internal deformation. This becomes evident by the fact that we see similar features to the ones we are prominently discussing for the FKIR, namely enhanced SMB on the windward side of the ice rise and erosion on the local ice divide, only in one of the three profiles. The one profile which also shows these features is the one parallel to the dominant wind direction. This is also additional evidence for the idea, that both processes generally occur on ice rises with ridges perpendicular to the predominant wind direction. We added a sentence to clarify this.
*L262-266: The absence of the local minima at the ice divide on profile 3 and 4 across the TIR (Fig. 3) provides further evidence that this is a wind driven accumulation feature and not a result of internal deformation. All three profiles are crossing one of the ridges of the ice rise at a similar elevation, however only the one parallel to the dominant wind direction (profile 2) shows the local SMB minimum and maximum near the peak of the profile (Fig. 3). Therefore, it seems that positive curvature of the topography, parallel to the dominant wind direction, is necessary to create this erosion and deposition feature.*

The implications of the work should be made more prominent. This work has conclusions that impact the interpretation of ice core records from ice rises, but these implications are only given a few sentences near the end of the discussion, and only just mentioned in the abstract and conclusions.

We agree with the referee that the implications are not prominent enough in the discussion. We changed lines 314-315 in the discussion and added a few sentences to highlight the implications. We also modified figure 7 to highlight the differences in SMB evolution between windward and leeward side. See below.

*L314-323: This becomes evident when looking at the temporal SMB evolution on the windward side of the ice rise compared to the SMB evolution at the peak of the ice rise or the leeward side (Fig. 7). While the SMB at the peak of the ice rise and on the leeward side decreases with time, the SMB on the windward side decreases only until 2002, but then increases in the latest time period from 2002 to 2018. This shows a disconnection between the SMB evolution at the peak of the ice rise (where the ice core is recovered) and the windward side of the ice rise, where snowfall is highest. Now since we observe high SMB on the windward side of an ice rise and erosion at the peak of an ice rise on other ice rises too, it is not unlikely that this disconnection between SMB evolution at the peak and on the windward side is also a generic ice rise feature. A consequence of this would be that an SMB record from an ice core at the peak of an ice rise alone would only be sufficient to derive the SMB at the very peak*

*of the ice rise and the leeward side, but would fail to capture the SMB evolution on the windward side of the same ice rise.*

1.3   Small, line-by-line stuff

Figure 1: wind-rose plot is difficult to read at magnifications under 200- suggest reworking that part of the plot if it's important, or eliminating it and replacing with a 'prevalent wind derection' arrow. The black dot at the summit of FKIR is not called out explicitly- presumably it is the ice core location but not clear in the caption- add it in.
Removed the wind rose and added a 'prevalent wind direction' arrow. Also added a note marking the black dot as the location of the ice core.

Line 27: hyphenate "inter-annual"
Changed to interannual by request of the other referee.

Line 52, set of references: this actually goes back to Black and Budd 1964 (who King et al cite)- might be good to cite them too! Black, H. P., and W. Budd (1964), Accumulation in the region of Wilkes,Wilkes Land, Antarctica,J. Glaciol.,5(37), 3–15.
Added the citation.

Line 62: Might as well name the ice rises before shortening them to abbreviations (I'm assuming the abbreviations are for "T" ice rise and "FK" ice rise).
FKIR and TIR were project internal names, named after leading scientist in the Mass2Ant field expedition to the ice rises. However since there are official norwegian names for both ice rises we decided to rename them. Therefore we changed FKIR to Lokeryggen ice rise (LIR) and TIR to Hammarryggen ice rise (HIR). But we kept the

name Derwael ice rise (DIR) since it was already named like this in earlier publications we are citing. We added a note though mentioning the norwegian name of DIR (Kilekollen).

Line 70: delete comma after "rises"
Done

Lines 76, 77: including make and model of the (probaby commercial) GPR would be of interest to GPR folks
Included the model of the GPR (GSSI:SIR 3000)

Figure 2: Fonts too small to read even at 135 zoom. enlarge. Also, the IRHs are challenging to see without magnification of close to 300. Can the contrast be improved, or a different color scheme be employed?
Increased font size and changed the color of the IRHs. See below

Line 102: Best not to start a sentence with an acronym, even if it's a common one like GPR.
Changed the sentence to:
*L102-103: To study the spatial SMB distribution across the ice rise we used a GPR. The GPR emits radar beams into the ground and records the time it takes for the signal to return, after being reflected on internal layers within the snowpack.*

Line 104: Niether of these references establishes that the IRH is an isorchron, and you really should include one (Callens in particular uses the vague "generally accepted" language that should be avoided. A good early reference for this is Spikes et. al., 2004 (see figure 2): Spikes, Vandy B., et al. "Variability in accumulation rates from GPR

profiling on the West Antarctic plateau." Annals of Glaciology 39 (2004): 238-244.
Added the citation.

Line 106: In section 2.1 you said you used 400 Mhz. Close enough that the science
doesn't change, but choose one (preferably the accurate one) and be consistent
(unless of course you used both!).
Corrected to 400 Mhz

Line 106: to clarify the language, suggest changing "the first 50 m of the snow- pack
in the vertical direction" to "the shallowest 50 m of the snowpack"
Done

Line 121: delete "do" from the phrase "in order to do get the density"
Done

Line 133: Not clear if the initial layer depth mentioned here is part of an iterative
procedure or simply part of an initial data analysis step. If the former, describe more
fully, if the latter, delete as it's not necessary!
Removed the unnecessary part of the sentence
*L132-133: This layer depth was calculated from the two-way travel time using a radar*
*velocity which increases with density following the mixing formula of Looyenga*

Line 135: with such detail in the procedure the omission of an actual equation describ-
ing mathematically how SMB is derived seems glaring. Suggest adding here.
We added an equation and some explanatory sentences.
*Finally we combined the density model, one layer depth value for every hundred meter*
*along GPR IRHs and the age dating from the ice core to calculate 220 SMB values for*

*each time interval, along the profile (Fig. 3 B). Each SMB was calculated by summing up the density between two IRH and dividing by their age difference.*

$$SMB = \frac{\sum_{z=d_i}^{d_{i+1}} \rho(z) * V_e}{A_{i+1} - A_i} \tag{1}$$

*Where $\rho(z)$ is the density at depth $z$ and $V_e = 1$ is a volume element in our case 1 m times 1 m times 1 m, $d_i$ is the depth of an IRH and $A_i$ is the age of that IRH. The error bars around the SMB profile represent the uncertainty due to the age measurement and the density model.*

Lines 138-148: This approach is fine, and using relative magnitudes of SMB is ok, but I don't think it's appropriate in that case to use absolute values on the y axes of Figures 3 D-F. instead, use some scaled value and leave units off. On line 146 you state that the absolute values should be disregarded- thus leaving them on the figure almost invites misuse of the result.
Removed the units from the y-axis

Line 161: "it consist of" -> "it consists of"
Done

Line 165: "for a, in our case, single. . . " awkward. Rewrite.
Changed it to:
*L165-166: SnowPack simulates the changes of snow depth for multiple snow layers, but does not consider any snow micro structure. Here we ran SnowPack with a single snow layer for simplicity reasons.*

Line 191: mention Figure 3 first, and then Figure 4.
Done

Figure 3: Again, because the SMB estimates in panels D-F are only relative, leave units off the y-axis.
Done. See below.

Figure 4: make the box in panel A (denoting the location of panel B) more prominent.
Done. See below.

Lines 203-204: start referring to figure 5 here; will be useful.
Done

Figure 5: Needs A and B labels. Also, either repeat legend in panel B (there's plenty of room), or move it outside both panels. Finally, since SMB in each model is what is being directly compared to the GPR SMB, make that curve bold.
Done. See below.

Figure 6: If a figure needs to be removed to save space, this one is a good candidate.
Fig. 6 is indeed the least important figure of all of them, but since we are claiming in the text that the arch amplitudes of the IRHs are increasing linearly with depth, we thought it might still be important to show this.

Lines 238 and 265: I don't see the ice rise DIR actually named anywhere.
Changed the text, DIR is now named in line 195 (Derwael ice rise)

Line 327: "For this ice rise the erosion at the peak. . . " This sentence is unclear- in particular the expression "ice rise wide temporal variations". I understand the idea but it is not well expressed here- suggest split to a few sentences. . .
Changed the sentence.
*L339-343: For this ice rise the erosion at the peak of the ice rise locally evens out the higher snowfall values on the windward side of the ice rise. Not only does the erosion at the peak reduce the higher snowfall values due to orographic uplift on the windward side, it also compensates for the higher temporal variability in snowfall on the windward side, as the erosion values are higher when more freshly fallen snow is available.*

————————————————

[Figure]

[Figure]

**Fig. 1.**

[Figure]

**Fig. 2.**

[Figure]

[Figure]

**Fig. 3.**

A)

B)

SMB in [mm w.e./yr]

**Fig. 4.**

**A)**

SnowModel

Legend:
- SMB
- Snowfall
- Wind redist.
- Sublimation
- SMB (GPR)

Y-axis: SMB in [m w.e./y]
X-axis: Distance to ice divide in [m]

**B)**

Racmo

Legend:
- SMB
- Snowfall
- Wind redist.
- Sublimation
- SMB (GPR)

Y-axis: SMB in [m w.e./y]
X-axis: Distance to ice divide in [m]

**Fig. 5.**

[Figure]

**Fig. 6.**

---

## Author Response (AR2)

**TC Response to editor**

Dear Editor,
Thanks a lot for the positive feedback. We will of course apply all of your suggested minor changes. As for your comment about Roosevelt island, that is actually really good to hear, because this far I have only seen observations of this in East Antarctica, and this shows that it is also happening in West Antarctica.

Best regards,
Thore Kausch and co-authors

**1    Editor comments:**

Comments to the Author:
The revisions look really good and do a great job of responding to the referees comments. Thanks!

I have just a few editorial comments, and one personal note. These corrections shouldn't take much time and are really quite minor. Again, ice job.

P11 L 40: "Their topography influences the surrounding SMB, by inducing snowfall, especially on the windward side of an ice rise,Xand blowing snow by alternating the wind speed patterns " : this sentence is a little confusing. Should it say something like "and by influencing the deposition of blowing snow..."
Done

P11 L55 "as well as " -> "or about"
Done

P12 L69: "them and how well" -> "them, and how well"
Done

P12 L70: comma after 'addition'
Done

P13 L104: "reflected on" -> "reflected from"
Done

P15 L156 Complicated sentence should be broken in half: "RACMO2 is a state of the art regional atmospheric climate model. Similar to, for example MAR (Agosta et al., 2019) and COSMO-CLM (Souverijns et al., 2019), it is able to provide accurate SMB simulations in polar regions.
Done

P16 L206: "the amount and contrasts " -> "the amount and the contrasts"
Done

P20 L252: linear->linearly
Done

: As it happens, I was on the trip to Roosevelt Island (at age 25, working on my Master's degree) and made similar measurements with a high-frequency radar. There was indeed an accumulation minimum just at the summit of the island. This has never been published (as far as I know), but I thought you might like to know.

**List of changes**

**2   General**

All changes were done as suggested by the editor.

**3   Data availability**

L358 changed "Data available on request from t.kausch@tudelft.nl and will be uploaded to a public repository with DOI on eventual final publication" to "Data available on request from t.kausch@tudelft.nl".

It is still the plan to upload the GPR data to a public repository. However, I removed that sentence here as I wasn't sure if I would still be able to change it before publication and I haven't uploaded the data yet. The reason for this is that I am currently unable to connect to the TU Delft data repository for some reason. Hopefully I will be able to fix this before final publication, and I am assuming I will still get another opportunity to include the DOI.

[revised manuscript text omitted]